# "Of course, drones delivering urgent medicines are necessary. But I would not use them until. . ." Insights from a qualitative study on users' needs and requirements regarding the use of medical drones

**Franziska Fink** [1,2], **Denny Paulicke** [1,3]*, **Martin Grünthal**[4], **Patrick Jahn**[1,2]

1 Health Service Research Working Group, Acute Care, Department of Internal Medicine, Faculty of Medicine, University Medicine Halle (Saale), Martin-Luther-University Halle-Wittenberg, Halle (Saale), Germany, 2 Faculty of Medicine, Martin-Luther-University Halle-Wittenberg, Halle (Saale), Translationsregion für digitalisierte Gesundheitsversorgung (TDG), Halle (Saale), Germany, 3 Akkon University of Human Sciences, Department of Medical Pedagogy, Berlin, Germany, 4 Pharmacy at the Bauhaus, Dessau-Roßlau, Germany

☯ These authors contributed equally to this work.
* denny.paulicke2@uk-halle.de

**Data Availability Statement:** All relevant data are within the paper and its Supporting Information files.

## Abstract

### Background

The current COVID-19 pandemic, demographic trends, and the increasing shortage of skilled workers pose major challenges for the care of people with and without care needs. The potential of drones as unmanned aerial vehicles in health care is being discussed as an effective innovative way of delivering much-needed medicines, especially in rural areas. Although the advantages are well known, the needs of the users have not yet been taken into account.

### Methods

Online-based focus groups (via *WebEx*) were conducted with participants from different disciplines: nursing, pharmacy, physicians. Focus groups with COVID-19 patients were conducted face-to-face. The focus was primarily on potential problems and requirements of the users regarding the use of drones. Structured and contrastive snowball sampling has been deployed. The focus groups were audio recorded, transcribed by a transcription-company, and coded with the help of the program "f4analyse 2" for content (Elo et al. 2008).

### Results

Especially during the pandemic situation, delays, and restrictions in the delivery of medicines have been noticed. All interview partners (patients, pharmacists, physicians, and nurses; n = 36 participants) see drones as useful in cases of limited mobility, time-critical medicines (rapid availability), emergencies, and disasters (e.g., floods), but also for the

**Funding:** This research was funded by the Federal Ministry of Education and Research Germany through the "TDG - Translational region of Digital Health Care" project [03COV25E]. PJ receives this funding. The funders did neither partake in study design, data collection, analysis, nor in the preparation of the manuscript or the decision to publish.

**Competing interests:** The authors have declared that no competing interests exist.

delivery of regular medicines in rural areas (e.g., for the treatment of chronic diseases). Moreover, only 16.7% of the participants have experiences with drones.

## Discussion

Drone deliveries do not play a role in the health system yet despite their great importance, which is perceived as particularly evident in the pandemic situation. The results lead to the conclusion that this is mainly due to knowledge and application deficits, so that educational and advisory work is absolutely necessary. There is also a need for further studies that go beyond the scope of acceptance research to describing and evaluating concrete scenarios of drone delivery on the basis of a user-centered approach.

## Introduction

With challenges in the health care system such as increasing urbanization/rural exodus, aging populations, and increasing shortages in health care workforce, it becomes evident that new, innovative solutions are needed to face these challenges. The infection control measures taken in the context of the COVID-19 crisis have exacerbated this effect and made it visible, since the elderly and people in need of care in this special situation are unable to provide themselves with urgently needed medicines [1, 2].

In the context of the ongoing COVID-19 pandemic, drones are discussed as a new logistic solution and hence are perceived as the most common application in health care [3]. In general, drones are widely used in medicine and health care in the areas of public health and disaster relief, telemedicine, and medical transport [4]. Drones in medical transport have been widely used for the delivery of medicines (e.g., vaccines, drugs), blood supplies, organs, defibrillators, and other medical supplies from one medical facility to another or to (COVID-19) patients [5, 6]. In this regard, the use of drones is a fast and contactless delivery strategy that can overcome large distances. Medical drones have an immense benefit to health care. They improve response times of emergency services, access to health services in difficult and remote areas, and clinical outcomes (e.g., survival after a cardiac attack) [3]. Moreover, medical drones are considered a more cost-effective alternative for air transport of medical supplies compared to helicopters in mountainous, desert, and forest areas where there is no access to roads or long distances, or in areas that have been affected by major natural disasters [7]. Although, the potential of drones in health care is considered high, technological, organizational, and environmental factors (TOE barriers) affect the use and performance of drones [3, 8]. Due to structural/organizational conditions and processes that have not yet been defined (legal conditions, framework conditions, etc.), the use of drones in Germany and many other countries is still limited to research projects and prototypes [3, 9]. The majority of studies were located in Africa due to more regulatory openness compared to Europe and Northern America [9, 10]. Nevertheless, studies conducted in Africa cannot simply be transferred to European contexts [11]. However, the most identified factor that affects drone use was the usability and design of drones [3]. In this vein, the lack of user's skills, negative perception about the technology as well as technology immaturity hinder the use of drones [8]. A positive attitude toward drones affects the willingness to use and interact with them for different purposes. Moreover, the technical interest in general and knowledge about drones are identified as supportive factors for acceptance [12]. According to the technology acceptance model (TAM), the attitude toward

using a technology is dependent on perceived ease of use and usefulness. In other words, better usability and higher usefulness of a technology lead to a higher acceptance of the same [13, 14].

## The present study

Even though, particularly in the COVID-19 pandemic, awareness of new and innovative forms of care has grown further [8], centering users' acceptance for medicine delivery with drones is little the focus of research. In a recent review, it was found that there are currently no studies that examine the interaction between humans and drones in the delivery of medicines [11]. The study also shows that there are hardly any approaches to explaining the needs and views of potential users of drones. However, the process of acceptance building starts before an initial contact with an innovation and continues into application phase [15]. Thus, the present study explores the question of what user requirements are in the delivery of medicines with drones under current pandemic conditions and beyond even before a drone-based delivery system has been developed. Understanding the context of users and their needs is an important step for designing solutions that are developed according to users' requirements [16, 17]. The present study is one part of a feasibility study, the pharmacy-drone-study (Apotheken-Drohnen-App; ADApp) that used a mixed-method design following the guidelines of co-creative user-centered design according to Farao et al. [18]. The concept of the overall project is provided in S1 Appendix. Against the backdrop of the current pandemic situation, scenarios for alternative solutions for the logistics of medications are being developed in a co-creative process as a part in the present study together with users, which can be taken over by the drone quickly and without contact, e.g., in the case of acute medication requirements in respiratory emergencies. This approach differs significantly from previous developments of technical solutions and drone app development, as the research is participatory from the beginning on and can thus focus more on acceptance and application of the app. This is also made clear by a systematic literature review conducted as part of the overall study, which confirms the research gap [11].

## Methods

### Study setting and sampling

As a part of the whole ADApp project, the present study investigates users' problems, needs, and requirements within the relevance cycle by conducting semi-structured discussions (focus groups; FG). The ADApp study was approved by the Ethics Committee of the Martin-Luther-University Halle-Wittenberg (protocol code 2021–069 and date of approval May 6, 2021). All participants gave written informed consent. Results of FG substantiate the further empirical procedure within the framework of the overall project. The discussion thereby pursued the goal of eliciting the subjective relevance frames of the participants in relation to the app and drone use to be developed. For improving the rigor, comprehensiveness, and credibility of reporting methods and results of the present FG study, we followed the COREQ Checklist [19]. The checklist is provided in S2 Appendix.

**Participants.** We selected participants representing the potential users of the drone-based medication delivery service, of the supply chain, and its technical components. Thus, participants were selected according to the role characteristics: general practitioners, nurses, pharmacists, and SARS-Cov2-infected-patients with rehabilitative treatment due to Long COVID condition resulting in four different FG. The FG were conducted separately and uniquely for each role characteristics. Participants were recruited with the support of the ADApp project team, the Paracelsus Harz-Clinics Bad Suderode, the Medical Journal (Deutsches Ärzteblatt), the Pharmacy-ADHOC Journal (Apotheke-Adhoc), and the DBfK—The German Nurses

Association (Deutscher Berufsverband für Pflegeberufe DBfK e.V.). Thus, the selection was purposive and followed the snowball-sampling. Participants were approached by telephone or via email. There was no relationship to participants prior to study commencement. All participants were informed about the procedure, the general aim, and reasons of the study and gave written informed consent before starting the discussion. A detailed description of the participants is given in the results section.

**Materials.**   The FG instrument (in German language) was developed from FS through a multi-stage systematic process. DP, MG, PJ, and members of the ADApp Team critically reflected the FG instrument in the context of a quality-assuring feedback [20]. An expert review is one potential pretesting method to determine whether the items of the FG instrument were problematic or clear in context of its goal. There was no specific instruction given to provide feedback on items, except for a short statement about an overall FG instrument goal [21]. After each feedback, the FG instrument was adapted. The FG instrument consisted of nine questions. These questions were based on the Technology Usage Inventory (TUI) [13, 14]. Three technology-specific factors (usability, usefulness, and accessibility) and four psychological factors (curiosity, interest, skepticism, and fearfulness) of the TUI were used for question development. The FG instrument includes questions about actual problems under pandemic conditions, the knowledge and competence in the handling with medication apps and drones, the general usefulness of a drone-based delivery, the usability features in communication, delivery process (and handover), the accessibility of the application and drone delivery, as well as concerns of using such technology. The interview guide and an overview of questions assigned to category and TUI factors is provided in S3 Appendix. Before starting the interview, two moderators (M1 and M2) introduced the participants to the topic and persons and provided a fictive case study.

**Data collection.**   The FG were conducted online via *WebEx* with exception of the patient group which was conducted during their rehabilitation stay at the Paracelsus Harz-Clinic Bad Suderode. During the FG discussion, two moderators (M1 and M2) guided the groups. Both moderators have introduced themselves by reporting their credentials and position in the ADApp project. M1 (first author: FS; PhD; research associate; female) controlled the communication process, provided the questions or prompts. The aim was to encourage the participants to discuss and to generate a creative atmosphere by taking up different ideas and opinions so that the participants can build up narrative ambitions through positive feedbacks [22, 23]. M2 (first author: DP; PhD; research associate; male or last author: PJ; PhD; Head of Working Group; male) simultaneously collected thoughts and aspects on an online flip chart (*Miro Board*) for orientation. These were taken up again later in the discussion. During the open narrative phases of the participants, M2 noted additional information of the interactions or the atmosphere of the discussion in order to obtain a later assignment to the data to be evaluated. It serves the purpose of exploration [24]. All moderators provide a high level of expertise and experience in qualitative methods whereby DP proves the most experience in conducting focus groups. Depending on the situational assessment, FG lasted between 60 and 120 minutes [23]. During FG discussion no one else was present besides the participants and the two moderators. At the end of the discussion, the participants had the possibility to leave further contact data in order to encourage participants for further participation and continuous monitoring ("core group").

**Data analyses.**   The notes taken by M2 were made available to participants and were discussed in the FG, so they could verify that their statements were noted without distortion. However, transcripts were not returned to participants. The participants' responses were audio recorded, transcribed by the company "abtipper", and coded with the help of the program "f4analyse 2". Due to the primary content-related objectives of the research interest and

**Table 1. Structured analysis matrix.** The category of usability was subdivided into communication, process, and handover.

| No. | Category | definition | example |
|---|---|---|---|
| 1 | *Problems* | describes current problems that justify the usefulness of the technology | "At the moment, we have a lot of isolated solutions that make us more work overall." |
| 2 | *usefulness* | describes the advantages, the benefits of the technology | "So, I think palliative care is of course an important point, for us and for the rural area and then of course elderly patients who need their basic medication." |
| 3 | **usability: general** | describes the necessities/conditions for the comprehensibility, application, and usability of the technology | "I would only use it if it is somehow even remotely compatible with my software." |
| 3.1 | *usability: communication* | describes the necessities/conditions for the communication (e.g., chat, video, call, etc.) of the technology | "But I think a quick answer is also important. Especially when you're now in quarantine, a bit on your own and maybe scared. If you need a medicine, you're usually not well and I think it's very important that you get an answer as quickly as possible." |
| 3.2 | *usability: process* | describes the necessities/conditions/characteristics of the process of technology | "There must be one drone launch per patient at the end of the day." |
| 3.3 | *usability: handover* | describes the necessities/conditions for the transfer of medication | "One of the requirements for the app would be a documentation that the delivery has actually been made. I think that's very important." |
| 4 | *accessibility* | describes the way of access to the technology, the related effort, and the necessary resources (technical, economic, financial, organizational) | "The prerequisite would then actually also be a wireless connection for the app." |
| 5 | *concerns* | describes the dangers, the risk, the complications, the disadvantages, the difficulties that the technology brings with it | "So, I think if someone has small children or dogs it's kind of difficult." |

the research question, a simple transcription system was used [25, 26]. Thus, only the manifest content of the transcripts were analyzed [27]. The data analysis was carried out according to the content analysis method from Elo and Kyngäs [27]. After becoming familiar with the transcripts by reading them several times, analyses were conducted using a deductive approach moving from the general to the specific. The unit of analysis was defined as words, one or several sentences that contain one theme and was coded in terms of a structured analysis matrix, i.e., only aspects that would fit the matrix were chosen from the data. The categories were prepared in tabular form, so that an independent evaluation was possible. FS coded transcripts for the categories. DP coded 25% of the transcripts independently and peer examination was conducted by the Health Service Research Working Group and Acute Care with extensive experience in qualitative research.

The data coding followed the categories underlying the FG instrument based on the TUI assessment [13, 14]: problems (i.e., curiosity and interest), usefulness, usability (split into communication, process, and handover), accessibility, and concerns (i.e., skepticism and fearfulness) (see **Table 1**). However, these categories were divided into different issues aiming at translating the main categories into a set of functional requirements and design guidelines relevant to the software system goals (see result section). The knowledge base has been queried with a short question of whether the participants have experience and knowledge about drones and medication apps.

## Results

### Participant's knowledge base

Between June 2021 and October 2021, we collected data from 36 participants in total. Participants were 44.4% female and 55.6% male, ranging in age from 23 to 75 years (mean $50.1 \pm 13.8$). According to the knowledge and competence in the handling with drones, only five participants out of 30 (16.7%) stated that they already have experiences with drones. General practitioners are having the most experience (100% out of three responses). Ten out of 31

Table 2. Participant characteristics (*n* = 36).

|  | general ractitioners | pharmacists | nurses | patients | total |
|---|---|---|---|---|---|
| *n* | 8 | 8 | 6 | 14 | 36 |
| 1. age | 47.5 (10.0) | 44.14 (9.6) | 39.5 (13.4) | 57.8 (14.2) | 50.1 (13.8) |
| 2. age range | 30–57 | 34–61 | 26–58 | 23–75 | 23–75 |
| 3. women (%) | 37.5 | 25 | 50 | 57.1 | 44.4 |
| 4. drone experience (%) | 100 | 14.3 | 0 | 7.1 | 16.7 |
| 4.1 number of answers | *n* = 3 | *n* = 7 | *n* = 6 | *n* = 14 | *n* = 30 |
| 5. medication app experience (%) | 75 | 85.7 | 16.7 | 0 | 32.2 |
| 5.1 number of answers | *n* = 4 | *n* = 7 | *n* = 6 | *n* = 14 | *n* = 31 |

Note: Numbers in brackets indicate standard deviation.

participants (32.2%) were familiar with the handling with medication apps, whereas pharmacists (85.7% out of seven responses) and general practitioners (75% out of four responses) are having the most experience. However, the patients indicated no competences in handling medication apps. Further results are reported in **Table 2**.

## Qualitative content analysis results

Different issues were extracted for five main categories. We identified five different problems (category 1), eleven issues of usefulness (category 2), seven issues of general usability (category 3), five of communication usability (category 3.1), six of process usability (category 3.2) as well as three of handover usability (category 3.3), ten issues of accessibility (category 4), and twenty different concerns (category 5). All issues were assigned to one of three factors, classified into technological, organizational, and economical (TOE) factors, influencing adoption of drones [8] (see **Table 3**). **Fig 1** shows the proportion of five main categories in relation to total utterances illustrated in percentage. It shows that the categories of usability (usability total) and concerns accounted for the majority of the discussions. General practitioners discussed most about within the usefulness category of drone-based medication delivery while pharmacists debated most about topics that match usability categories. Within the usability subcategories, the handover was the most debated usability issue. Nurses conversed most about possible concerns and difficulties involved in the drone delivery. However, the concerns had the most discussion compared to the other categories. The most important examples are presented below. More detailed and in-depth examples are provided in S4 Appendix.

**Problems.** COVID-19-patients experienced no support during their disease or during the pandemic situation, especially in rural areas. *"But when you've had Covid, I was in bed for four weeks, the 5th week they did a little gymnastics with me. When I came home, I could barely walk 5 meters, that's how tired I was. I only gradually reacquired that with my own training. Nobody supports you, no hospital, especially if you live in a rural village."* However, pharmacists stated that a drone for COVID-19 aftercare would have been useful. Furthermore, patients and pharmacists commented that the supply in rural areas is limited. This is mainly due to the fact that young people move to the city and the older ones stay in the village. Some can no longer drive a car and are therefore dependent on the care of others which is not only an issue during pandemic situations: *"There are 30 people living in our village. Out of the 30 people, 12 are over 80, then we have about 13 people who are over 60, and then we actually have only a handful of young people. Almost everyone has an own house, the children have moved out. For them the idea is very good with drone delivery, that's really nice I must say. The oldest in our village is 95, he can't drive a car anymore, and he always has to ransom someone from the family to do it."*

**Table 3. Users' issues assigned to TOE factors per category.**

| | organizational factors | environmental factors | technical factors |
|---|---|---|---|
| **knowledgebase** | • lack of users' expertise about drones and medication apps | | |
| **problems** | • delayed / limited medication supply | • limited support<br>• pandemic regulations<br>• economic efficiency | • digital isolated solution |
| **usefulness** | • limited mobility<br>• rapid medication availability<br>• emergency and disaster<br>• regular medications<br>• reliefs<br>• retention reductions<br>• logistic reduction<br>• delivery at night<br>• flexibility | • fewer cars and papers<br>• contactless delivery | |
| **usability: general** | • speech recognition<br>• regular medications<br>• data sovereignty and security<br>• control over delivery process (plannability) | | • additional device<br>• interface integration<br>• (only automated solutions without analog steps)<br>• scalable drones for different use cases |
| **usability: communication** | • consultation options<br>• reconciliation options<br>• medication plan and intake information<br>• note functions for delivery | • drone labeling | |
| **usability: process** | • shipment tracking (including delivery location and navigation)<br>• bilateralism<br>• order summary<br>• infrastructure (adaptable to daily routines) | | • photo function for physical prescriptions<br>• availability and charging status of drones |
| **usability: handover** | • identification<br>• payment via app | | • winch or landing |
| **accessibility** | • information through pharmacists and general practitioners, broad media, health insurance companies<br>• space for landing<br>• need for general information and education/training<br>• corporations<br>• user-centered approaches and pilot testing | • economic sense<br>• competition advantage through consulting via app<br>• uniform regulations<br>• political and general practitioner commitment | • network ac• cess |
| **concerns** | • stigmatization through visible delivery<br>• lack of humanity<br>• lack of users' digital competences and fine-motoric skills<br>• communication barriers<br>• app over floating<br>• lack of users' experience and knowledge about drones<br>• space for drones<br>• handover in case of immobility and apartment house<br>• lack of accessibility<br>• identification during handover | • lack of economy<br>• lack of regulations<br>• loss, crashes, and injuries with drones<br>• drone destruction<br>• regulation uncertainty (e.g., flying over people's properties, insurances. anesthetics, theft)<br>• misuse<br>• ownership | • poor network access, especially in rural areas<br>• lack of resistance to climate conditions<br>• drones' noise |

(patients). The pandemic situation is not only having an impact on patients. General practitioners and nurses are reporting restrictions to office hours that result in a decrease in treatments which in turn leads to delayed diagnoses. Moreover, the general practitioners said that it was difficult for them to stay in contact with the patients without physical contact which was prone to errors. They *"[. . .] gave the prescription by mail. And if there had been a mistake, the patient would have had to report it. But we couldn't recheck that now."* They also reported delay in medication delivery due to the pandemic situation. Pharmacists emphasized the increasing palliative care, where rapid delivery is necessary. Nevertheless, they reported *"[. . .] we have a*

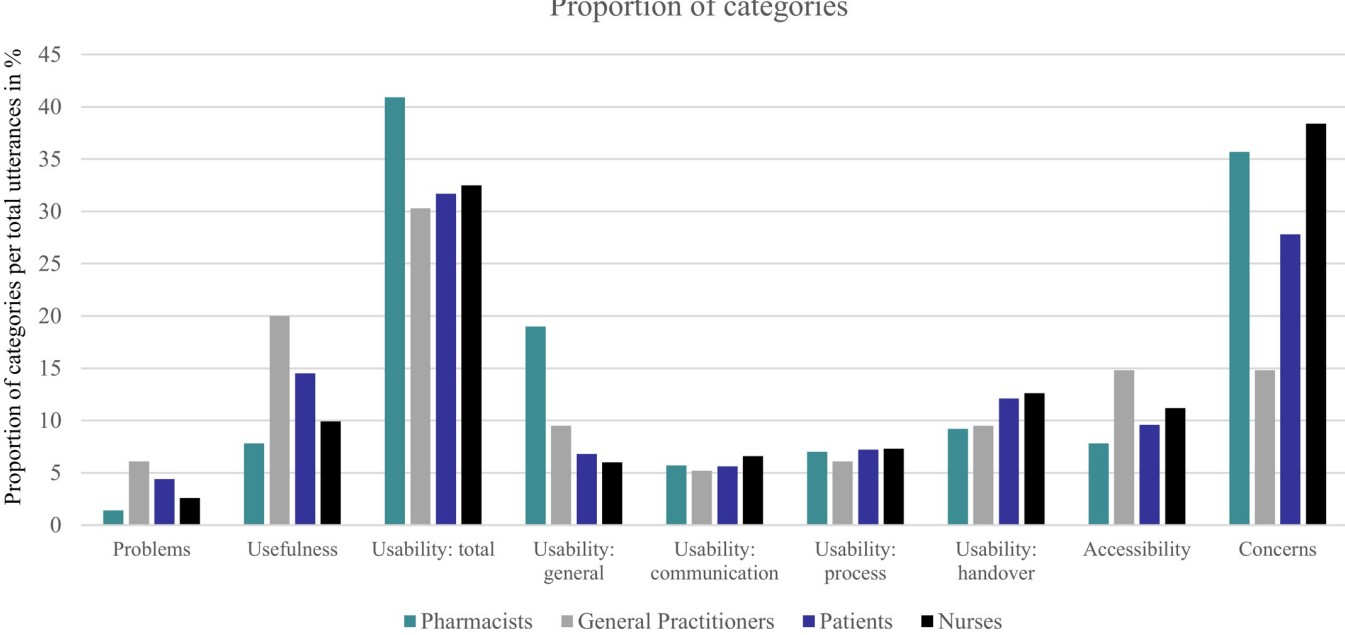

**Fig 1. Proportion of categories.** Proportion is per total utterances for pharmacists, general practitioners, nurses, and COVID-19-patients for the categories: problems, usefulness, usability, accessibility, and concerns. Usability is subdivided into general, communication, process, and handover. All usability subcategories are merged into usability total.

*lot of isolated solutions that are making more work overall."* Additionally, they talked about the lack of economic efficiency of digital solutions which however could facilitate their work.

**Usefulness.** Fitting to the problems of rural care and the lack of support during Covid-19, all sample groups agreed that a drone-based medication delivery is useful in cases of immobility or limited mobility. They mentioned for example the supply of rural areas or areas with poor (pharmacy) infrastructure, long distances, and heavy traffic or single/fewer mobile people who have no connection to care by the outpatient care service, e.g., in case of quarantine or after surgeries. General practitioners and nurses strengthened this by a rapid availability and delivery of time-critical medicines to patients at home or to clinics and outpatient care service. However, patients and general practitioners also noted the importance of the delivery aspect. They find it useful in cases of emergency (e.g., pandemic situations) and disaster (e.g., flooding) and emphasized the advantage of contactless delivery. In addition, they felt that it would be important for drone-based medication delivery to be demand-driven. They suggested to include the delivery of regular medicines to patients and institutions (e.g., in the care of chronic diseases).

Independent of the pandemic or emergency situations, the participants indicated the usefulness on the relief for pharmacies, patients, and general practitioners (patients), the support for local pharmacists (patients) through an increase in attractiveness (pharmacists), the support for environment (e.g., fewer cars (patients)), the reduction of medication retention in clinics (general practitioners), the delivery of medical and laboratory products or blood (general practitioners), and the reduction of logistical processes (general practitioners). Pharmacists mentioned that the service allows them to be more flexible and innovative. This innovative character was expanded by the idea of *"[. . .] delivery at night. Because I was often called: "Can't you bring it over to me? I'm a 90-year-old granny. I can't get out of there." That's why I would use it at night."*

**General usability.** This category includes the subcategories "communication," "process," and "handover". However, seven issues are identified as usability that cannot be assigned to the subcategories. In situations where people are in need, immobile, or have fewer digital competences, patients and nurses proposed speech recognition possibilities. Thus, the ordering medications and the announcement according to the delivery process could be facilitated. Furthermore, patients suggested an additional device so that not everything is smartphone-based which would accommodate people with low digital competences. *"If I'm in a situation where I'm in need of help, then such an additional device can really be a blessing. I think everyone has to weigh it up for themselves. I could imagine that a message comes through such a device: "the medication is on the way or that is then there and that is deposited there, in front of the door, on the terrace."."*

When talking about usability, issues outside the pandemic and emergency scenario were also raised. Patients and nurses reiterated the emphasis on need-based delivery that delivery should also be extended to regular medications (e.g., chronic diseases). Especially nurses said that the medications should be delivered to the outpatient care office. Thus, the app should indicate when the outpatient care service has office hours during which the medication can be delivered, and general practitioners should be able to see in the app which nursing service is responsible for a certain patient. *"I imagine if the emergency general practitioner writes a prescription at night, he can see that nursing service XY is responsible for this patient and would then perhaps send the prescription to the responsible nursing service so that the nursing service can bring the medication in the morning."* General practitioners and pharmacists talked about the interface integration, i.e., they only would use it if the app was integrated into practice software, web-based with password access or QR code (general practitioners), or provided an interface integration with pharmacy merchandise management systems (pharmacists). The pharmacists emphasized that they would not tolerate a stand-alone solution outside pharmacy technology otherwise it would be more work and therefore unattractive for pharmacists: *"So, I would only use it if it is somehow even remotely compatible with my software."* They also indicated that there should be only digital and automated solutions: *"But when we talk about a project like this, it actually has to be completely digital. We can't have any analog steps in between."*

For patients it is important that the user has the data sovereignty and is able to delete contact data themselves while general practitioners talked about app security, i.e., that the app should be without errors. Moreover, general practitioners like to have control over the process and the medication. That means, the general practitioners would like to release the medication for the drone-based delivery depending on whether a patient is able to receive it by drone. Pharmacists prefer to have an own drone for more plannability. Furthermore, pharmacists and general practitioners discussed about different drones for different use cases. They suggested scalable drones depending on range, flight speed, and payload (i.e., small drones for urban and rural areas for short distances and individual customers as well as large drones for urban and rural arears for institutions and clinics).

*Communication Usability.* All participants thought the app needed to have consultation options between general practitioners and patients as well as between pharmacists and patients, e.g., via video call, chat function, call option, or feedback form, especially when a person is in need or quarantine. In this vein, pharmacists confirm that perspective: *"Counseling comes first and foremost. We are health care professionals. We want the patient to come first, not our economic interests. [. . .] similar to tele-pharmacy."* Additionally, nurses find it important that they themselves and patients get information about the medication plan (i.e., instructions of what is important with other medications) and medication intake in form of digital instruction leaflets: *"I can imagine that the app might also provide instructions on how to take the medication. I don't know if everyone in the outpatient care service is always familiar with all the medications, how*

*they should perhaps be taken, and instructions perhaps also for the patients themselves."* Moreover, they also would like to have note functions in the app for delivery notes.

Independent of pandemic situations, patients prefer that drones would be labeled as medical drone for increasing the social acceptance when a drone flies over another property. General practitioners prefer to have partner pharmacies for callback and coordination opportunities: *"Of course, you have to coordinate with the pharmacies beforehand to find out which ones will participate. Because the system only works if I have at least one partner pharmacy that converts the e-prescription into a real medication in the physical world."*

*Process usability.* All sample groups expressed a general necessity of shipment tracking. They would like to have a fixed and transparent process, i.e., information about the process status, fixed contact persons, and fixed delivery times. Nevertheless, patients noted that they want information limited to a minimum: the request is in process and the drone is arriving at a certain time. They also stated the wish of forwarding the e-prescription from general practitioner directly to pharmacy making the process as lean as possible. The app should display the delivery locations when living in a city and should navigate to the next drop-off location (nurses): *"That the app maybe displays where the next delivery locations are now."* Nurses also like to have an indication of the last deliveries in an order summary. They also find it useful to include the possibility to send the medications to the neighbor. For patients and pharmacists, it is important that the operations and processes are as simple as possible. *"When I'm really sick as a patient I should be able to use it. That it is extremely easy to handle purely in terms of function."* (patients).

Bilateralism represents another issue of process usability which was addressed by general practitioners and nurses. Bilateralism means that either the general practitioner issues an e-prescription, or the patient requests an e-prescription (for example in the case of chronic diseases), i.e., the possibility to request a medication and e-prescription from the general practitioner because a medication is needed, or the supply is running low. *"Maybe it goes there, in reality, that you can also request something. That you can trigger that as a nursing service or say, here the supply is exhausted, and that then the corresponding general practitioner writes the prescription."* Nurses also discussed about a photo function in the app, so that the nurse who *"[. . .] comes to the patient, who just has a visit from the general practitioner now and has a prescription lying there [. . .]"* can *"[. . .] scan it and sent it. Maybe also, so for the next day, with us for delivery, or, if it is urgent, for the evening additionally."* However, pharmacists have completely different issues for process usability. They talked about the issue of infrastructure. Especially pharmacists indicated that it is necessary that the app is adaptable to the daily pharmacy routine: *"We always have a pharmaceutical staff checking it. And then it's delivered by a non-pharmaceutical staff. And we can do it the same way with the drone."* They expressed the need of drone loading areas that can be reached from more than one pharmacy and the need of physical conditions in pharmacy, e.g., charging possibilities. They also discussed about the availability of the drones. For them it is a requirement that the availability and charging status of the drone is visible, e.g., within the app, so that it is clear whether the drone is ready for use or not.

*Handover usability.* The issue of handover usability most discussed by all FG was the identification during medication receipt. The participants discussed about different possibilities of identification, e.g., through fingerprint, voice recognition, code entry, electronic signature, images, token exchange, PIN, TAN, or QR code. Even if general practitioners know about a certain residual risk because they *"[. . .] never know one hundred percent what will happen to the anesthetics prescription if* [they] *give it out somehow"*, they commented that *"[. . .] it should be technically controllable"* and possible. Another issue discussed was the form of handover via winch (patients, nurses, general practitioners) or landing (patients, pharmacists). The winch

was indicated as mainly useful in urban areas (patients) while landing seems to be a useful scenario in rural areas (patients, pharmacists) or at institutions with a central landing area (patients, nurses, general practitioners). The general practitioners came up with the idea of *"[. . .] placing a kind of target QR code somewhere and the drone recognizes with the camera that there is a QR code—there is "my landing location" [. . .].* The pharmacists came up with the idea to deliver the medicine in rural areas to a local central point, e.g., the local lottery store. For urban areas, nurses generated a few ideas for the landing scenario: landing on the roof, landing at a landing pad that can be opened from the outside of buildings or gripper arms at the drone which passes the medication through the window. Patients imagined that the drone could hover and release the medicine with a winch in front of the window so that the patient only needs to open the window and take the medication. Nevertheless, in case of landing, the location must be accessible to the patients or relatives (nurses). However, landing or winch are both scenarios that were accepted by all participants in general. A smaller issue that was discussed was the method of payment. Only patients talked about that issue and said that they would like to pay within the app or by invoice.

Taken together, the main issues discussed by all participants were consulting (communication usability), shipment tracking (process usability), and identification during handover (handover usability). **Fig 2** shows the proportion of three main usability issues per total utterances. It becomes visible that shipment tracking was taken up a lot of discussion space within the patient and nurses sample groups. While the identification during handover was less discussed for patients, it was the biggest issue for pharmacists. Moreover, another big issue a lot discussed by pharmacists and also general practitioners was the counseling.

**Accessibility.** In response to the question what information channels they would use to learn about such technologies, the participants talked about four different issues: the access to the technology and the necessary resources related to environment, organization, and technique. With regard to the access to technology, pharmacists want to inform patients and nurses what has also been reflected by patients and nurses. Patients and nurses also would like to have information about broad media such as newspaper, television, social media, or flyer. Moreover, patients suggested cooperation with health insurance companies so that the information about drone-based deliveries also reaches them. Last but not least, patients think that information spreads by word of mouth, especially in rural areas. However, general practitioners want their patients to be informed either by pharmacies or via the mass media, but above all they want to make recommendations themselves. Therefore, general practitioners want to be informed about the drone-based medication delivery first through general practitioner-specific publications (e.g., medical journal, medical congresses, or through the ADApp project itself). Pharmacists in turn prefer an active marketing by themselves and drone manufactures.

However, participants also expressed necessary resources that come along with the drone-based medication delivery. Pharmacists and patients talked about environmental and economic factors. Pharmacists *"[. . .] need a higher benefit for higher costs."* They debated about when the delivery makes economic sense, but they also agreed that a drone-based delivery is useful if medicines *"[. . .] have a certain significance."* But not all pharmacists agreed with the view on economic factors. Although for all the statements it was important that *"[. . .] consulting comes first."*, some declared that *"[. . .] patients come first, not our economic interests."*. However, they also talked about another economic difference: that consulting is also a competition advantage because *"[. . .] that´s the difference between what [mail-order pharmacies] don ´t understand. With them, it´s all about the price.* The patients wondered *"[. . .] how much it will be used."* But they also commented that *"the more it is used, the cheaper it becomes. [. . .] and the more common it is, the higher the acceptance will be."* Patients also want to have uniform regulations otherwise it is more confusing than meaningful when all general practitioners

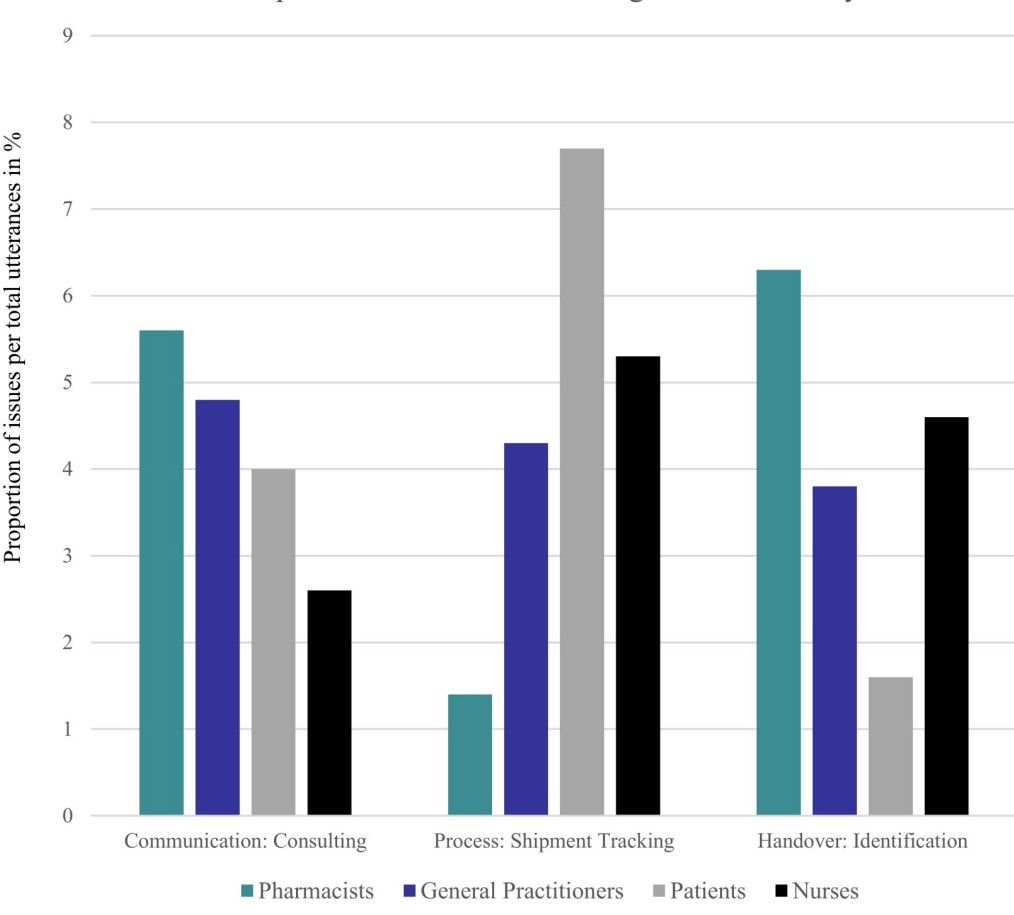

**Fig 2. The proportion of issues of usability categories per total utterances.**

and pharmacies do it differently. Political commitment and general practitioner commitment seems indispensable for pharmacists and patients: *"You have to convince general practitioners of that, too, because otherwise it doesn't work."* (patients). Patients also raised another important issue: the technical requirements for using such technology. They stated out the problem of network access in rural areas. But especially network access and Wi-Fi are important prerequisites for the use of the app. Next to these requirements, all FG discussed about organizational resources such as enough space for the drone to land (patients). The patients also noted that general information and education to people is necessary so that the *"[. . .] people see the sense behind it, what is behind such drones."* Another organizational issue is that in case of immobility cooperation with nursing services are required for supporting the handover. Thus, nurses or at least relatives must be at the patient's house so that the patient receives the medicines (patients, nurses). Moreover, there is a requirement of cooperation between pharmacists and outpatient care service (nurses). However, general practitioners like to have more field reports respectively pilot projects that test for functionality in reality, which in turn would show necessities and would increase their general trust. Pharmacists seem to prefer a user-centered approach that works from bottom to up: *"The reason I like this initiative is because it comes from the pharmacy community, from research, and not directly from the top. Anything that comes from the top is then not quite realistic."*

**Concerns.** Participants consistently expressed feelings and concerns when talking about the drone-based medication delivery. Seven issues have been discussed which were not only triggered by the closing questions but occurred constantly in the course of the FG. A big issue highly discussed by all sample groups was the human aspect. Within the patient FG, the issue of their COVID-19 disease was most visible when talking about concerns of drone-based medication delivery. Some of the patients felt stigmatized during their disease through that may stand in connection with a disease. This was also an issue when they talked about the drone-based delivery. They wondered *"what if I am really sick and everyone around me is healthy? And now someone sees that I'm being supplied with a drone there, you're also stigmatized."* In this vein, they also are afraid of data misuse which has an impact on their feeling of security, especially when having an infection.

A fertile debate was carried out about the human component that some of the patients and general practitioners mentioned as a deficiency. *"Do you know what it was about? It was about feeling that someone was helping her. This human component. . .I also have a cell phone and I do a lot on the Internet and also I am ordering things, but I'm missing a bit of humanity."* (patients). The general practitioners discussed the missing human component as well. They meant that *"[. . .] it's important that the medicines get to the patient. I know that patients are very grateful when there is a person handing over the meds, that is, a human being at the door and not some drone landing in front of the building"* (general practitioners).

All participants agreed that the lack of digital competence, especially for older people, could be one of the biggest problems. *"I think it's difficult. Many patients don't even have an iPhone or a smartphone. So, they can't handle the technology"* (general practitioners). General practitioners also noted that when drugs are very fragile, they *"[. . .] wouldn't rely on it being picked up reliably* [by patients]*",* which was also a matter of concern for the nurses. Another important point nurses mentioned was that *"the older people are no longer fine motored. How do they get to these medicines well?"* Pharmacists believe that *"the customer won't take one more app to do it, too".* Nurses confirm this assumption by saying that medical apps sometimes are too complicated in managing such a huge amount of prescriptions. Patients also confirmed this by discussing about the focus on cell phones. However, pharmacists also said: *"I don't think it matters how* [the patients] *get* [the medicine]*."* In contrast, patients thought about the impact on work conditions and were concerned about the loss of jobs through drones.

All participants also stated that *"[. . .] most people don't have enough experience with drones"* (patients). They talked about their own inexperience as well as the general lack of sufficient knowledge about drones including the capacity of the load, quantity of medications, charging, flight range, and landing procedure. This was particularly evident in the following commentary: *"Our pharmacies drive really fast and are very accessible and by phone we have good communication"* (general practitioners). Additionally, nurses wondered about *"what medicine is urgent, but not so urgent that the emergency general practitioner has that with him? And if it's really urgent, then maybe he will take the patient with him to the hospital."*

An issue that also all participants discussed was the economical aspect. *"Because of the economics, I can imagine that the use cases where that's really limited to those emergencies are pretty small and that probably then tends to lead to pharmacies maybe not taking the system because it's just too expensive"* (patients). *"It's also a question of what it costs afterwards"* (general practitioners). Nurses also noted that also other *"[. . .] companies have been involved with the delivery of parcels, for example.".*

In addition to human or economic related aspects, all sample groups also talked about organizational and infrastructural aspects. They expressed skeptics and ambiguity about patients' involvement in processes, interactions in cities, and the level of patient participation in communication. Pharmacists have concerns about the space in and the location of their

pharmacies: *"What do I do when I have an airport nearby?"* Patients talked much about their concerns of how to get their medicine if they live in the city and/or are immobile. They are also concerned about the possibility of the medicines being stolen in the city. Nurses cannot imagine how medicines can be delivered in an apartment house in the city and reached by patients, especially in case of immobility, too. *"Or if you just live in an area where it's difficult to impossible to get that done with the drone"* (general practitioners). Nurses discussed most about the implementation of drone in the ambulatory nursing service: *"The handover, I consider this to be a problem, with us at least, in the ambulatory area."* However, the patient focus laid on the problem of poor or no access to network and Wi-Fi, mainly in rural areas.

All sample groups talked also about legal aspects: *"If it fails, it will fail on that"* (pharmacists). When talking about this issue, participants wondered about the loss or crashes of the drones (pharmacists, patients), the rights of flying over other people's properties (patients) as well as insurances (pharmacists). Generally, pharmacists, nurses, and general practitioners were afraid of delivering anesthetics because *"[. . .] that is already dangerous. Carrying with me 3000, 4000, 5000 euros I would not want to drive out with the drone at the current state"* (pharmacists). General practitioners additionally noted *"If someone else gets the drone, then it's difficult."* Nurses wondered *"[. . .] how is the shipment protected from unauthorized unloading?"* Besides these concerns, pharmacists wondered if they need a shipping / trading permission when delivering medicines by drone. In a broader context, general practitioners are concerned that the clarification of airspace will become too complex.

Further, the participants considered drone-related factors. Fitting the law framework, pharmacists, nurses, and patients are concerned that the drone could injure patients, children, and dogs. In addition to questions about the rights of flying over other people's property, general practitioners and patients said this increases the risk that drones could be destroyed. *"There are drone haters who, after all, don't know that a person is waiting for their medicine right now"* (patients). Pharmacists, nurses, and patients have concerns about the reliability of delivery due to darkness or weather dependence of drones. Moreover, patients find it *"[. . .] a great solution that you get something at all, but I don't know if that can be slowed down once it's up and running. That it will take on a life of its own, that other providers or companies will take it up and I don't know if you can stop it so that there really isn't drone traffic in the air all of a sudden."* This matches the patients' worries about the noise of drones and the impairment caused by drones for people and animals. Pharmacists also have completely different questions about the availability of the drones and its ownership which has implications for their daily work. Another issue pharmacists talked about concerns accessibility to the special service. They noticed that the drone-based medication delivery *"[. . .] would be a nice story, but the difficulty is really getting through."*

Despite many proposed solutions, a major issue the pharmacists struggled with was the documentation of the identification of the patients when receiving the delivery. *"I've also thought about how I actually want to document that the person, the patient, has accepted it. That's the most difficult thing, I think, isn't it?"*

## Discussion

The application of drones in the supply process of medication is generally supported by the participants of this focus group discussion. They described a differenced picture of their expectation regarding human drone interaction and the implementation in the app-based supply chain.

Despite the enormous focus and benefit of drones in health care, and especially in the delivery of medical supplies [3], it is surprising that drones are not yet part of the health care system. Besides legal conditions, the most important factor affecting the use of drones is user acceptance [3]. To date, reviews and studies focus on social acceptance of civil drones or the benefit

and challenges of drones in society [3, 4, 8, 9, 12]. Using "user-centered-designs" is one important design to develop user acceptance positively and to further increase acceptance of drones in health care settings. Thus, the present study is embedded in a mixed-methods study design and is intended to provide needs and requirements of users in a specific scenario of drone-based medication delivery amid the ongoing COVID-19-pandemic situation.

Generally, the present study shows problems of perceived limited support and delays in medication in the context of the pandemic situation. In this vein, drones are considered useful in cases of limited mobility, time-critical medicines (rapid availability), or of emergency (e.g., pandemic situations) and disaster situations (e.g., flooding), but also for the delivery of regular medications (e.g., in the care of chronic diseases). However, the study demonstrates that very few of the respondents had any knowledge and competence about drones (16.7%) or medication apps (32.2%). According to results of Eißfeld et al. [12], a general technical interest and knowledge about drones play an important role for acceptance. In other words, the better people are informed about benefits and challenges of drones, the more they accept them. It can be concluded that concerns and usability aspects, especially the handover, accounted for the majority of discussions over all focus groups of all five categories (problems, usefulness, usability, accessibility, and concerns). With regard to the low level of knowledge about drones and medication apps, it is not surprising that concerns make up for the most discussion [12]. However, these results imply the need for general information and education to increase technical knowledge and competence. According to the TAM, the attitude toward using a technology is dependent on perceived ease of use and perceived usefulness [13, 14]. Thus, not only general clarifications of drones are necessary, rather providing drone technologies and systems for a specific scenario in health care is able to shape experience in a positive way. The ongoing ADApp study will provide insights whether the iterative involvement of users during technology development will increase user acceptance. At this point, responses of users can be implemented in the present development of an app-assisted drone-based medication delivery.

### Issues influencing user acceptance

Organizational, environmental, and technical factors are considered to affect the use and performance of drones whereby usability and design of drones affect them most [3, 8]. According to these TOE factors, the present study identified needs and requirements of users, which can each be assigned to these TOE factors (see **Table 3**). The frequency of discussed issues shows a certain heightening of users´ needs. Just in one review it was shown that it became apparent that usability aspects contribute to the most proportion of discussion which is also present in our results [3].

**App-related issues.**   The communication aspect seems to be more important when people are in need. In a previous study using an AED delivery scenario, it was found that interaction with the dispatcher gave the participants a sense of security that results in a positive effect on compressions [28] (see **Table 4** for a comparison between results of a previous scoping review [11] and the present study). Thus, one important issue influencing the acceptance of drones in the delivery of medicines is the need of consultation features via app. It seems that when humanity decreases, technologies in turn need arrangements to meet the need of more relatedness [29]. These results show the benefit of user-centered studies since the consulting aspect has been often overlooked so far [11]. Another important usability aspect regarding communication is that all information and the entire process have to be continuously digital without analog steps in between. Especially the pharmacists emphasized that they would not tolerate a stand-alone solution outside pharmacy technology otherwise it makes more work, and it would be not attractive

Table 4. Comparison of results of previous scoping review [11] and present study.

| Scoping Review [11] | Present Study | | Scoping Review [11] | | Present Study |
|---|---|---|---|---|---|
| Perceived Support | Usability | | Concerns and Wishes | | Concerns |
| *app-related* | | | | | |
| interaction with dispatcher | consultation | | | | |
| live-video streaming | identification handover | | | | |
| *drone-related* | | | | | |
| | | | injury by drone | = | injury by drone |
| | | | lack of resistance to climate conditions | = | lack of resistance to climate conditions |
| | shipment tracking, delivery status | | uncertainty about drone's arrival | | |
| | drone labeling as medical drone | | reflective tapes or lights for marking location of arriving | | drone destruction |

for them. This implies that new technologies involving different parties have to be integrated in and connected with existing health care services in an automated manner (IoMT).

Another important organizational factor that has to be included in a pharmacy-drone-app is a shipment tracking for a transparent delivery process. These issues were also identified in a previous scoping review [11] (see **Table 4**). It was found that users expressed uncertainty and anxiety about the lack of knowledge of the direction the drone comes from, the drone's look and sound, as well as the landing and handover process in an AED delivery scenario [28, 30, 31]. This was the most identified response of users (measured in terms of the number of studies) [11]. Thus, users need the information about the delivery process for reducing uncertainty and concerns, which is especially useful in situations where people are in need. From a basic psychological perspective taking needs into account, these results can be assigned to an agglomeration of autonomy, relatedness, and competence, i.e., the need of having meaningful choices in an effective manner while feeling connected to others [29]. Sufficient information about the delivery process could make the user feeling safe, connected, and able in handling the process. It provides a feeling of agency, overall. Another important issue discussed was the point of bilateralism, i.e., either the general practitioners make out an e-prescription or the patient requests an e-prescription. Thus, an app like the pharmacy-drone-app should include features of a bidirectional process that simplifies the process. This might be useful in case of chronic diseases to facilitate accessibility to patients and to reduce costs. In this sense, there is a need for finding a balance between the accesses to new technology for the most vulnerable and maintaining of economic interests.

Regarding handover, the most discussed organizational factor was the identification during the receipt of the medicine. Especially pharmacists discussed the most about the handover process. They expressed concerns about the manner of documentation that the medicine has actually been delivered / received, which is most important when transporting anesthetics.

However, participants expressed a high discrepancy between the usefulness of drones, especially in rural areas, and the lack of network connectivity in rural areas, digital literacy, and fine-motoric skills, especially when getting older. Patients and nurses suggested speech recognition features within the app to order the medicine in a voice-controlled manner and to get an announcement according to delivery status, which makes it easier for people with less digital literacy or impairments. Above all, patients were upset about the fact that every technology can only be operated with smartphone, producing hence an app flood and that in case of loss all data are gone. However, beside a cloud-based solution, patients suggested an additional device (homedot) like an Alexa dot or Homepod but only for medical matters. Another

discrepancy is that drones are useful in cases of limited mobility but not useful in case of immobility when a patient cannot leave the house or live in a region where it is difficult to impossible to get the medication by drone (e.g., in cities). Regarding these concerns, participants expressed different ideas. One important issue is the coordination between the ambulatory nursing service and the drone-based delivery so that either the medication would be delivered to the office of nursing service or during the visit at patient's house.

**Drone-related issues.** A big issue highly discussed by patients was their experiences of stigmatization during COVID-19 disease. They expressed concerns about the visible and noisy delivery of medications with the drone. Controversially, they talked about risks that drones could be destroyed by other people who do not appreciate drones flying over their properties and possibly being unaware about the importance of the drone. However, patients prefer to label drones as medical drones since the arrival of medicines in emergencies seems more important than the stigmatization. Moreover, the technical development of drones should be directed toward reducing noise. The stigmatization and noise pollution were expressed as concern. A recent study of Eißfeld et al. [12] revealed that concerns about noise have the strongest impact on acceptance. However, results of the present study showed that participants believe that acceptance of drones would increase when the service of drone-based delivery will become more common, when more user-centered approaches and field reports respectively pilot projects are performed that test for functionality under real life conditions. The participants mentioned that such projects show necessities and increase their general trust because they are also concerned about the abuse of drones to criminal purposes. Another study has also shown that participants mentioned such concerns (91%) [12]. Thus, more user-centered research on drones with concrete scenarios are necessary to improve acceptance and trust.

Studies have shown that the physical contact with the drone is one of the biggest user concerns [28, 30, 31]. One solution could be using a winch that reduces direct physical contact with the drone. Results of the present study also suggest that participants prefer the handover of medications via winch, although landing was a discussed alternative, which is implementable in rural areas or at institutions with a central landing area (see **Table 4**).

However, some other factors influencing the performance of drones in health care are concerns about the lack of the drone's resistance to climate conditions, especially at night. In the future, drones should be developed to be more resistant to harsh conditions. Lights at the drone could be a solution for at night deliveries [11]. However, patients need the commitment that the medication is getting to them as fast as possible. Above all, legal restrictions and questions of insurances make it difficult to test drones in real context. That is the reason why pilot projects are conducted mostly in Africa. There is clearly a need of studies in Europe to investigate the role and acceptance of drones in European health care because only in this way laws may be changed and adapted and the adaption from low and middle income countries in Africa is very limited, but the economic perspective is truly better in the health care systems of the high income countries To reach this the first important step is, that users be aware of solutions to their problems and should accept new technologies before superordinate structures can be changed. Moreover, health workers must learn about how to integrate drones into health supply chains by involving them into the development of such technologies. However, this could be a fact that is of no difference regarding Africa and Germany since a case study found that users expressed that donors and operators "do what they want, and not what we want" [10].

## Limitations

When interpreting the results, several methodological limitations have to be taken into account. The sample corresponds to the typical comparative figures with regard to age and

gender distribution, but cannot be classified as representative due to the small number of participants. However, due to the first saturation of the data material, a certain generalizability of the results can be assumed [24]. The participating actors from the health sector had a fundamental interest in new topics and in the issue itself. However, the risk of a possibly "positive selection" is not detected, as the topic area of drones and their use for medical deliveries was mostly unknown to the health care actors. Thus, positions that consistently reject drones in the context of medical supply delivery were just as little represented as those that refrained from participating in the focus group for other reasons. Reasons for this could be, among others, a general negative attitude toward additional efforts due to time resources, heavy workloads, or other thematic priorities. Nevertheless, it can be assumed that the results and interpretative derivations of the present study could, under certain conditions, also be valid for other parts of Germany as well as for other areas in which the use of drones in health care is in focus (e.g., in palliative care). Regarding the linguistic aspect of the contribution, it should be mentioned that all focus groups were conducted and documented in German. Even though the quotations used in this paper were translated into English with the help of a native speaker, a literal reproduction is not always given since accessibility and readability are of importance, too. Any slight shift in meaning was checked using back-and-forth-translations.

## Conclusion

As this study illustrates, COVID-19-patients, pharmacists, nurses, and general practitioners think that drones offer an alternative option for transporting medicines in order to help people in need to improve access to medical supply, especially in rural areas and for people with limited mobility. However, they described the lack of economic efficiency of digital solutions and the increase in isolated solutions. At this point, connecting an app with a drone system seems useful for participants to know the location of handover and to adjust the medicine receipt. Thus, participants emphasized that the pharmacy-drone-app must be as lean and simple as possible (e.g., with speech recognition and medical voice-controlled homedots), must facilitate processes and routines (e.g., software integration, plannability, shipment tracking), has to be completely digital, must enable control (e.g., data sovereignty, handover documentation), and should include consultation and reconciliation features. There is clearly a need of user-centered studies investigating the role and acceptance of drones in health care to learn about how drones can be integrates into the health supply chain system. User-centered designs have the added value that users find the solutions to their problems on their own in exchange with each other and thus have a higher impact in technical implementation. The participatory development of technical solutions together with the users proves to be positive with regard to acceptance and integration of care and should be further expanded.

## Supporting information

**S1 Appendix. ADApp design.**
(DOCX)

**S2 Appendix. COREQ checklist.**
(DOCX)

**S3 Appendix. Interview guide.**
(DOCX)

**S4 Appendix. In-depth examples.**
(DOCX)

## Acknowledgments

We would like to thank the ADApp-Team for exchange and discussion of the focus group guide and for evaluating the results of focus groups for further technical development. A special thank goes out to Prof. Dr. Axel Schlitt from the Paracelsus Harz-Clinics Bad Suderode, the Medical Journal (Deutsches Ärzteblatt), the Pharmacy-ADHOC Journal (Apotheke-Adhoc), and the DBfK—The German Nurses Association (Deutscher Berufsverband für Pflegeberufe DBfK e.V.) for their help with participant recruitment. We would also like to thank Dr. Anne-Marie Lachmund who was responsible for proof-reading and stylistic adjustments.

## Author Contributions

**Conceptualization:** Franziska Fink, Denny Paulicke, Patrick Jahn.

**Formal analysis:** Franziska Fink, Denny Paulicke, Martin Grünthal.

**Funding acquisition:** Patrick Jahn.

**Investigation:** Franziska Fink, Denny Paulicke, Patrick Jahn.

**Methodology:** Franziska Fink, Denny Paulicke, Patrick Jahn.

**Project administration:** Patrick Jahn.

**Supervision:** Patrick Jahn.

**Writing – original draft:** Franziska Fink, Denny Paulicke.

**Writing – review & editing:** Patrick Jahn.

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
