## [Decision Letter · Decision Letter 0]

10 Oct 2022

PONE-D-22-13044“Of course, drones delivering urgent medicines are necessary. But I would not use them until…” Insights from a qualitative study on users’ needs and requirements regarding the use of medical dronesPLOS ONE

Dear Dr. Paulicke,

Thank you for submitting your manuscript to PLOS ONE. After careful consideration, we feel that it has merit but does not fully meet PLOS ONE’s publication criteria as it currently stands. Therefore, we invite you to submit a revised version of the manuscript that addresses the points raised during the review process.

We look forward to receiving your revised manuscript.

Kind regards,

Chi-Hua Chen, Ph.D.

Academic Editor

PLOS ONE

2. PLOS requires an ORCID iD for the corresponding author in Editorial Manager on papers submitted after December 6th, 2016. Please ensure that you have an ORCID iD and that it is validated in Editorial Manager. To do this, go to ‘Update my Information’ (in the upper left-hand corner of the main menu), and click on the Fetch/Validate link next to the ORCID field. This will take you to the ORCID site and allow you to create a new iD or authenticate a pre-existing iD in Editorial Manager. Please see the following video for instructions on linking an ORCID iD to your Editorial Manager account: https://www.youtube.com/watch?v=_xcclfuvtxQ.

3. One of the noted authors is a group or consortium [Denny Paulicke]. In addition to naming the author group, please list the individual authors and affiliations within this group in the acknowledgments section of your manuscript. Please also indicate clearly a lead author for this group along with a contact email address.

Reviewers' comments:

Reviewer's Responses to Questions

**Comments to the Author**

1. Is the manuscript technically sound, and do the data support the conclusions?

Reviewer #1: Yes

2. Has the statistical analysis been performed appropriately and rigorously? 

Reviewer #1: Yes

3. Have the authors made all data underlying the findings in their manuscript fully available?

Reviewer #1: Yes

4. Is the manuscript presented in an intelligible fashion and written in standard English?

Reviewer #1: No

5. Review Comments to the Author

Reviewer #1: Thank you for the opportunity to review this paper. It is a very interesting study and the findings will have significant implications in this current era. While most sections are very well written in terms of scientific arguments, several sections are too long and should be made more concise so as not to put off the readers. Especially certain sections in the Results - please try and trim some of the quotes and also limit the number of quotes used. Consider perhaps using a tabular format to capture and present some of the quotes. Please find the rest of my comments appended.

6. PLOS authors have the option to publish the peer review history of their article (what does this mean?). If published, this will include your full peer review and any attached files.

Reviewer #1: No

---

## [Author Response · Author response to Decision Letter 0]

16 Nov 2022

Dear Reviewer,

Thank you very much for the opportunity to submit a revised version of our manuscript. We are very thankful for the constructive comments provided by the reviewers. In the following, we address all reviewers´ comments in a point-by-point manner and hope the manuscript is now ready for publication in PLOS ONE. Changed paragraphs were marked in blue color throughout the manuscript. 

Thank you very much.

Yours sincerely,

Denny Paulicke (corresponding author), on behalf of all co-authors

---

## [Decision Letter · Decision Letter 1]

20 Feb 2023

PONE-D-22-13044R1“Of course, drones delivering urgent medicines are necessary. But I would not use them until…” Insights from a qualitative study on users’ needs and requirements regarding the use of medical dronesPLOS ONE

Dear Dr. Paulicke,

Thank you for submitting your manuscript to PLOS ONE. After careful consideration, we feel that it has merit but does not fully meet PLOS ONE’s publication criteria as it currently stands. Therefore, we invite you to submit a revised version of the manuscript that addresses the points raised during the review process.

We look forward to receiving your revised manuscript.

Kind regards,

Anand Nayyar, Ph.D.

Academic Editor

PLOS ONE

Journal Requirements:

Additional Editor Comments:

The Paper needs revisions and is subject for re-review.

Reviewers' comments:

Reviewer's Responses to Questions

**Comments to the Author**

1. If the authors have adequately addressed your comments raised in a previous round of review and you feel that this manuscript is now acceptable for publication, you may indicate that here to bypass the “Comments to the Author” section, enter your conflict of interest statement in the “Confidential to Editor” section, and submit your "Accept" recommendation.

Reviewer #2: (No Response)

Reviewer #3: All comments have been addressed

2. Is the manuscript technically sound, and do the data support the conclusions?

Reviewer #2: Yes

Reviewer #3: Yes

3. Has the statistical analysis been performed appropriately and rigorously? 

Reviewer #2: Yes

Reviewer #3: Yes

4. Have the authors made all data underlying the findings in their manuscript fully available?

Reviewer #2: Yes

Reviewer #3: Yes

5. Is the manuscript presented in an intelligible fashion and written in standard English?

Reviewer #2: Yes

Reviewer #3: Yes

6. Review Comments to the Author

Reviewer #2: Dear Authors, congratulations!

After a long analysis, it was observed that all requested corrections were successfully met.

From this perspective, the present article is approved to be published in this excellent journal.

Grateful.

Reviewer #3: A manuscript with a very attractive title. Abstract very well designed. The work in the introduction lacks information about what is novelty in the work, what gap it fills. In the research part, the authors imprecisely describe the participants in the research. This section needs to be refined, lines 118-130. The authors write that the participants were selected in the process but are not definitively characterized. It is also not known how many there were or on what basis the selection was made. The literature review in the work is quite poor and requires refinement. The conclusion also needs some fine-tuning.

7. PLOS authors have the option to publish the peer review history of their article (what does this mean?). If published, this will include your full peer review and any attached files.

Reviewer #2: **Yes: **Gabriel Gomes de Oliveira

Reviewer #3: No

---

## [Author Response · Author response to Decision Letter 1]

29 Mar 2023

Dear Reviewer,

Thank you very much for the opportunity to submit a revised version of our manuscript. We are very thankful for the constructive comments provided by the reviewers. In the following, we address all reviewers´ comments in a point-by-point manner and hope the manuscript is now ready for publication in PLOS ONE. Changed paragraphs were marked in green color throughout the manuscript. 

Thank you very much.

Yours sincerely,

Denny Paulicke (corresponding author), on behalf of all co-authors

---

## [Decision Letter · Decision Letter 2]

24 Apr 2023

“Of course, drones delivering urgent medicines are necessary. But I would not use them until…” Insights from a qualitative study on users’ needs and requirements regarding the use of medical drones

PONE-D-22-13044R2

Dear Dr. Paulicke,

We’re pleased to inform you that your manuscript has been judged scientifically suitable for publication and will be formally accepted for publication once it meets all outstanding technical requirements.

Kind regards,

Sathishkumar V E

Academic Editor

PLOS ONE

Comments from Editorial Office: Please note that the requests from Reviewer 3 regarding novelty are not a requirement for publication, as this is not part of the publication criteria at PLOS ONE.

Additional Editor Comments (optional):

Reviewers' comments:

Reviewer's Responses to Questions

**Comments to the Author**

1. If the authors have adequately addressed your comments raised in a previous round of review and you feel that this manuscript is now acceptable for publication, you may indicate that here to bypass the “Comments to the Author” section, enter your conflict of interest statement in the “Confidential to Editor” section, and submit your "Accept" recommendation.

Reviewer #2: All comments have been addressed

Reviewer #3: All comments have been addressed

2. Is the manuscript technically sound, and do the data support the conclusions?

Reviewer #2: Yes

Reviewer #3: Yes

3. Has the statistical analysis been performed appropriately and rigorously? 

Reviewer #2: Yes

Reviewer #3: Yes

4. Have the authors made all data underlying the findings in their manuscript fully available?

Reviewer #2: Yes

Reviewer #3: Yes

5. Is the manuscript presented in an intelligible fashion and written in standard English?

Reviewer #2: Yes

Reviewer #3: Yes

6. Review Comments to the Author

Reviewer #2: Congratulations dear authors, after a long analysis it was observed that all corrections were successfully carried out. Therefore, I approve of the current work.

Reviewer #3: Authors write in line 108; "This approach differs significantly from previous developments..."... I recommend that the authors refer to specific studies whether their work differs from other publications. It still lacks emphasis on what is novelty at work. Conclusion requires refinement from the side of the presentation, which was done in the literature on the subject.

7. PLOS authors have the option to publish the peer review history of their article (what does this mean?). If published, this will include your full peer review and any attached files.

Reviewer #2: **Yes: **Gabriel Gomes de Oliveira

Reviewer #3: No

---

## [Editor Report · Acceptance letter]

28 Apr 2023

PONE-D-22-13044R2 

“Of course, drones delivering urgent medicines are necessary. But I would not use them until…” Insights from a qualitative study on users’ needs and requirements regarding the use of medical drones 

Dear Dr. Paulicke:

I'm pleased to inform you that your manuscript has been deemed suitable for publication in PLOS ONE. Congratulations! Your manuscript is now with our production department. 

Kind regards, 

on behalf of

Dr. Sathishkumar V E 

Academic Editor

PLOS ONE